# Health Risk Assessment of Inhalation Exposure to Airborne Particle-Bound Nitrated Polycyclic Aromatic Hydrocarbons in Urban and Suburban Areas of South China

**DOI:** 10.3390/ijerph192315536

**Published:** 2022-11-23

**Authors:** Peng Gao, Feng Deng, Wei-Shan Chen, Yi-Jia Zhong, Xiao-Lu Cai, Wen-Min Ma, Jian Hu, Shu-Ran Feng

**Affiliations:** 1Institute of Architecture and Engineering, Guangzhou Panyu Polytechnic, Guangzhou 511483, China; 2Tianjin Key Laboratory of Water Resources and Environment, School of Geographic and Environmental Sciences, Tianjin Normal University, Tianjin 300387, China; 3Research Center for Eco-Environmental Sciences, Chinese Academy of Sciences, Beijing 100085, China; 4School of Business, Hong Kong Baptist University, Hongkong 999077, China

**Keywords:** PM_2.5_, TSP, nitro-PAH, inhalation exposure, health risk

## Abstract

Airborne particulates (PM_2.5_ and TSP) were collected from outdoor and indoor areas at urban (Haizhu District) and suburban (Huadu District) sites from 2019 to 2020 in Guangzhou. Three nitro-polycyclic aromatic hydrocarbons (nitro-PAHs) in the airborne particulates were identified by a gas chromatograph equipped with a triple-quadrupole mass spectrometer. In the Haizhu District and Huadu District, the nitro-PAH concentrations in PM_2.5_ and TSP did not show a significant decrease from winter to summer. From 2019 to 2020, the difference in the average concentration of nitro-PAHs in PM_2.5_ and TSP in Guangzhou was relatively low and had no statistical significance. The diagnostic ratios of 2-nitrofluorene (2-NF)/1-nitropyrene (1-NP) in TSP are less than five, while for 2-NF/1-NP in outdoor PM_2.5_ in the summer of 2019 and 2020 are more than five, which indicates that nitro-PAHs in the atmospheric PM_2.5_ in Guangzhou during summer mainly originated from the secondary formation of atmospheric photochemical reactions between parent PAHs and oxidants (·OH, NO_3_, and O_3_). 9-Nitroanthracene (9-NT) made the most significant contribution to the total nitro-PAH concentration. The incremental lifetime cancer risks (ILCRs) of nitro-PAHs in PM_2.5_ and TSP by inhalation exposure indicated low potential health risks in the urban-suburban of Guangzhou.

## 1. Introduction

Industrialization and urbanization have resulted in an increasing level of fine particulate matter, or particulates, with a diameter of less than 2.5 μm (PM_2.5_) [1,2,3]. Numerous studies have found that PM_2.5_-bound chemical components, such as polycyclic aromatic hydrocarbons (PAHs) and their derivatives, can cause harm to humans [4,5,6]. Polycyclic aromatic hydrocarbons (PAHs) are regularly monitored and have been widely studied in the literature due to their carcinogenic and mutagenic potential, while PAH derivatives such as nitrated PAHs (nitro-PAHs) are not included in existing air monitoring programs and have only recently attracted attention in the context of urban air quality [4,5,6,7]. 

Nitro-PAHs exist in both the gas and particle phases in the atmosphere; their gas/particulate partition depends on factors such as the vapor pressure, temperature, and the concentration and properties of the particle. Nitro-PAHs are ubiquitous environmental pollutants that mainly originate from imperfect combustion and the pyrolysis of organic matter [6,7]. Several studies have reported another formation pattern of nitro-PAHs, which was through gas phase reactions between parent-PAHs and oxidant radicals, such as OH, NO_3_^−^, and O_3_ [8,9,10]. In the atmosphere, it is widely assumed that OH or NO_3_ radicals attack the C atoms on the aromatic rings in the PAH molecules, followed by an addition of NO_2_ to form the OH-PAH or NO_3_-PAH adducts at the *ortho* position, and a loss of water or nitric acid to nitro-PAHs. The OH-PAH-NO_2_ loses H_2_O via a four-membered transition state to form nitro-PAH, and the NO_3_-PAH-NO_2_ adduct loses HNO_3_ via a six-membered transition state to form nitro-PAH [9,10,11,12].

Over the last few years, nitro-PAHs have become a source of special concern, since they are more hazardous than PAHs due to their mutagenic, genotoxic, and carcinogenic characteristics. 1-nitropyrene (1-NP) is categorized in Group 2A (probably carcinogenic to humans) [9]. The mutagenicity and carcinogenicity of nitro-PAHs are high compared to those of their parent PAHs because they promote the formation of reactive oxygen species (ROS), some of which are strong mutagens and carcinogens [13]. Mutagenicity has been documented for fluoranthene, pyrene, chrysene, benzanthrone, and their nitrated derivatives; developmental toxicity has been documented for phenanthrene, anthracene, pyrene, benzanthrone, and their nitrated derivatives [14,15]. The environmental sources of nitro-PAHs include diesel emissions, coal-fired emissions, gas fuels, liquefied petroleum gas, etc., especially the combustion process of fossil fuels [16,17,18] and the reaction of PAHs with hydroxyl radicals, ozone, and nitrogen oxides in the atmosphere [19,20]. Zhao et al. reported that vehicle emissions were one of the primary sources of nitro-PAHs. After the traffic policy was implemented, the nitro-PAH concentration was reduced to 22.49 pg/m^3^ in Langfang, Hebei Province [21]. Karavalakis et al. pointed out that the usage of biodiesel led to a decrease but also an increase in several nitro-PAHs concentrations in the emissions [22]. Albinet et al. pointed out that incomplete combustion of coal/biomass would release nitro-PAHs as well [23]. According to the previous literature [17,18,24], the concentrations of nitro-PAHs in emissions from combustion units (simulated, diesel, or gasoline vehicles and power plants) were usually 1–2 orders of magnitude lower than those of their related PAHs. Some studies reported that the concentrations of nitro-PAHs emitted from diesel engines are much higher than those from gasoline engines [25,26].

In urban areas, the concentrations of total nitro-PAHs are typically in the range of several pg/m^3^ to a few ng/m^3^, although the number of compounds included in each study differs, leading to additional variance [25,26,27,28,29,30,31,32]. Only a limited amount of information on the nitro-PAH baseline concentration can be obtained in urban cities because of local emissions. Therefore, further studies should be conducted at background sites with less anthropogenic pollution and fewer local pollution sources to evaluate the distribution and biological effects of these particles in less urbanized areas compared to urban areas. The aim of this study is to determine the spatial and temporal variation in nitro-PAH concentrations in fine particles in indoor and outdoor areas of Guangzhou, ranging from urban to suburban areas. The long-range transport potential of these pollutants is addressed by determining the particulate mass fraction and the mass size distribution. 

## 2. Materials and Methods

### 2.1. Sample Collection

The indoor and outdoor environments of Haizhu District (urban area) and Huadu District (suburban area) were chosen as sampling points for PM_2.5_ and TSP in this study. A total of 208 samples and 16 field blanks in sampling points were simultaneously collected during winter (December 2019 and 2020) and summer (July to August 2019 and 2020). PM_2.5_ and TSP samples were, respectively, collected continuously using high-volume samplers (1000 L/min, Wuhan Tianhong Company, Wuhan, China) in outdoor areas and using medium-volume samplers (100 L/min, Qingdao Laoying Environment Technology Company, Qingdao, China) in indoor areas for 24 h. Ambient meteorological parameters (including temperature, relative humidity, and wind speed) and pollution parameters (NO_2_, SO_2_, and O_3_) were monitored in the sampling site of the Haizhu District. Before sampling, the quartz fiber filters were prebaked in a muffle furnace for 6 h at 600 °C to eliminate organic interference and then equilibrated under constant temperature (25 °C) and humidity (55 °C) for 24 h. The filters were put into a refrigerator, sealed with aluminum foil, and preserved at −20 °C. After sample collection, the quartz filters were usually wrapped in aluminum foil and stored at −20 °C until analysis.

### 2.2. Sample Extraction and Analysis

Three nitro-PAHs were examined in this study: [9-nitroanthracene (9-NT), 2-nitrofluorene (2-NF), and 1-nitropyrene (1-NP)]. The pretreatment of filters was similar to that in previous studies [32,33,34]. Briefly, the quartz fiber filters (PM_2.5_ and TSP) were cut into several pieces. Naphthalene-d^8^ and phenanthrene-d^10^ were added to the samples as internal standards before extraction. Each filter was extracted twice using pressurized liquid extraction with hexane and acetone (1:1, *v*/*v*). For chemical analysis, the extracts were purified and eluted into vials. The analysis procedure for the 3 nitro-PAHs from the PM_2.5_/TSP filter samples is described in the Appendix A.

### 2.3. Quality Control and Assurance

All glassware used during the sample extraction procedure was rinsed with ultra-pure water and then baked at 100 °C to dryness. During the sampling period, blanks were collected to evaluate any background contamination and the results showed that PAH derivatives were not detected in these blanks. The detection limits of the method were 0.0192–0.513 pg/m^3^ for nitro-PAHs. The recoveries were 87–118% for nitro-PAHs. The recoveries of nitro-PAH surrogates were determined. Before sample extraction and analysis, substitutes were added to the extraction cells to monitor the whole analysis process. The recoveries of 2 spiked perdeuterated surrogates were 87–98% for naphthalene-d^8^ and 94–123% for phenanthrene-d^10^. The process blank, laboratory blank, and field blank were analyzed, and the measured values of nitro-PAHs could be ignored.

### 2.4. Exposure Assessment

The respiratory pathway is the main method by which nitro-PAHs enter the human body. The various components of nitro-PAHs have different toxicities, with differing risks of harm to the human respiratory system. Benzo[a]pyrene (BaP) has been identified as a strong carcinogen. Previous studies measured a more reasonable and accurate ratio of the carcinogenic toxicity of nitro-PAHs to BaP through toxicological tests and proposed the concept of toxic equivalence factors (TEFs) [35,36]. This study referred to the relative study results of TEFs of nitro-PAHs and calculated the total equivalent carcinogenic concentration (BaP equivalent concentration, BaP_eq_) of 3 kinds of nitro-PAHs with BaP as the reference. This aimed to assess the health risk of nitro-PAH pollution in PM_2.5_ and TSP. The calculation method is shown in Formula (1).
BaP_eq_ = ∑ *C_i_* × *TEF_i_*(1)
BaP_eq_ is the total equivalent carcinogenic concentration with BaP as the reference, ng/m^3^; *C* is the concentration of the *i*-th nitro-PAHs, ng/m^3^; *TEF* is the carcinogenic equivalent factor of the *i*-th nitro-PAHs. Carcinogenic and mutagenic equivalent factors of nitro-PAHs are shown in Appendix A.

The average daily dose (ADD) refers to the daily intake of nitro-PAHs via the respiratory pathway. This study mainly considers the exposure dose of nitro-PAHs through respiratory exposure and the lifetime cancer risk of the population due to nitro-PAHs. Due to differences in ages, sexes, physiological characteristics, and lifestyle habits, the population was divided into children, adult males, and adult females for this study. The ADD of respiratory exposure pathways by different people can be calculated by Formula (2):ADD = (*C* × IR × F × EF × ED)/(BW × AT)(2)
ADD is the average daily exposure of toxic substances in PM_2.5_ and TSP mg/(kg·d); *C* is the concentration of toxic substances (e.g., nitro-PAHs) in PM_2.5_ and TSP. This study is based on the BaP total equivalent carcinogenic concentration of the bioavailable concentration released by toxic substances in simulated human lung fluid (ng/m^3^); IR is respiratory volume m^3^/d; EF is the exposure frequency (days/years); ED is exposure duration (years); F is the indoor and outdoor stay time; BW is body weight (kg); AT is the average exposure time (days). The exposure parameter is the key factor in determining the accuracy of the environmental health risk assessment. The closer the selected exposure parameter is to the actual situation of the target population, the more authentic the results of the environmental health risk assessment will be. According to the manual of the Chinese population exposure parameters issued by the Ministry of Environmental Protection of China at the end of 2013 combined with the actual situation of Guangzhou, the population exposure evaluation parameters are determined. The specific parameters are shown in Appendix A.

### 2.5. Health Risk Assessment

According to the US EPA’s health risk assessment model for respiratory exposure to carcinogenic and toxic substances, the incremental life cancer risk (ILCR) of nitro-PAHs in PM_2.5_ and TSP [20,37] is as follows:ILCR = q × ADD(3)
q is the human carcinogenic intensity coefficient calculated according to animal experimental data, kg·d/mg; ADD is the average daily dose, mg/(kg·d). According to the US EPA IRIS, the carcinogenic intensity coefficient (q) of BaP through respiratory exposure is 3.14 kg·d/mg. The US EPA proposes that the level of incremental lifetime cancer risk (ILCR) of pollutants is acceptable between 10^−4^–10^−6^; that is, the exposure level is acceptable when the number of health hazards or deaths caused by exposure to pollutants is not more than 1 per 10,000 to 1 million people. An ILCR lower than 10^−6^ indicates that the risk degree is not significant, and generally, the necessity of risk management is not great; an ILCR of 10^−6^–10^−5^ indicates that a risk is higher than the risk of daily activities and deserves attention; an ILCR of 10^−5^–10^−4^ indicates that close attention needs to be paid and a solution should be adopted; an ILCR higher than 10^−4^ indicates that the risk degree is significant, which requires the adoption risk management to treat.

### 2.6. Statistical Analysis

The PM_2.5_- and TSP-bound nitro-PAH concentrations and profiles were compared using IBM SPSS Statistics 20.0 software, with a statistical significance of *p* < 0.05.

## 3. Results and Discussion

### 3.1. Nitro-PAHs in PM_2.5_ and TSP

In the summer and winter seasons of 2019, the average concentration of PM_2.5_ outdoors in the Haizhu district of Guangzhou was 24.7 ± 9.12 and 66.5 ± 18.8 μg/m^3^, respectively, and in Huadu District was 28.6 ± 6.60 and 69.6 ± 12.0 μg/m^3^; Figure 1 shows that the 35.0 μg/m^3^ objective PM_2.5_ concentration (GB 3095–2012) was exceeded in both locations in winter. TSP refers to particulate matter suspended in the air with an aerodynamic diameter less than or equal to 100 μm. In summer and winter, the average concentration of outdoor TSP in the Haizhu District of Guangzhou was 41.5 ± 13.8 and 117 ± 41.1 μg/m^3^, respectively, and was 48.8 ± 12.2 and 132 ± 29.2 μg/m^3^ in Huadu District. The average concentration of TSP in both locations was under the 300 μg/m^3^ limit set by the state air quality standard (GB 3095–2012) (Figure 1).

The temporal and spatial variations in nitro-PAH concentrations in PM_2.5_ and TSP in Guangzhou are shown in Table 1 and Table 2. In the summer of 2019, the average nitro-PAH concentrations outdoors and indoors in PM_2.5_ in Haizhu District were 0.116 ± 0.0125 and 0.0855 ± 0.0321 ng/m^3^, respectively, and nitro-PAHs in TSP were 1.11 ± 0.242 and 0.797 ± 0.173 ng/m^3^. In the winter of 2019, the average nitro-PAH concentrations in PM_2.5_ in Haizhu District were 0.195 ± 0.0176 (outdoors) and 0.152 ± 0.0230 (indoors) ng/m^3^, and nitro-PAHs in TSP were 1.34 ± 0.202 and 0.996 ± 0.0643 ng/m^3^, respectively. In the summer of 2019, the average nitro-PAH concentrations outdoors and indoors in PM_2.5_ in Huadu District were 0.447 ± 0.267 and 0.169 ± 0.0290 ng/m^3^, respectively, and nitro-PAHs in TSP were 0.892 ± 0.130 and 0.642 ± 0.225 ng/m^3^. In winter 2019, the average nitro-PAH concentrations in PM_2.5_ in Huadu District were 0.235 ± 0.0245 and 0.177 ± 0.0341 ng/m^3^, respectively, and the nitro-PAHs in TSP were 0.712 ± 0.192 and 0.462 ± 0.0837 ng/m^3^. There were differences between the average concentrations of nitro-PAHs in PM_2.5_ and TSP from different districts of Guangzhou but the differences were not statistically significant (*p* > 0.05). Owing to the low latitude, the surface of Guangzhou receives more solar radiation. Meanwhile, it is affected by the monsoons, the climate features high temperatures, high humidity, and much rain in summer, while in winter, the climate features low temperatures, dryness, and little rain. In the Haizhu District and Huadu District, there was no decrease in the average nitro-PAH concentrations in PM_2.5_ and TSP from dry winter to wet summer. The small direct emission of nitro-PAHs in summer and photochemical reactions may be important causes of this pattern. The urban air pollution in Guangzhou in summer is mainly O_3_ and NOx pollution [37,38], which leads to the secondary formation of various nitro-PAHs when exposed to free radicals. In nitro-PAHs, the formation of 1-NP is most significant when exposed to NO_3_/N_2_O_5_ and ·OH radicals, while 9-NT can be formed when exposed to NO_3_/N_2_O_5_ radicals. Therefore, the concentration levels of nitro-PAHs in PM_2.5_ and TSP do not significantly decrease in the wet summer, which may be related to the formation of photochemical reactions with exposure to free radicals.

The source of nitro-PAHs can be identified by using the marker method, such as that using 1-NP as a “marker” of nitro-PAHs derived from diesel emissions and using its existence in the surrounding air samples as a sign of traffic pollution from diesel vehicles. Other compounds directly emitted from the combustion chamber of the diesel engine are 9-NT and 2-NF. This study investigated the spatial-temporal contribution of pollution sources of nitro-PAHs in Guangzhou by analyzing the ratio of 2-NF and 1-NP (2-NF/1-NP). A ratio of 2-NF/1-NP less than five indicates that the primary emission source is dominant, and a ratio greater than five indicates that the nitro-PAHs are produced by secondary pollutants [39,40]. The ratio of 2-NF/1-NP in indoor and outdoor PM_2.5_ and TSP in the Haizhu and Huadu districts of Guangzhou from 2019 to 2020 is shown in Figure 2. 2-NF/1-NP in TSP was less than five, while 2-NF/1-NP in outdoor PM_2.5_ in the summer of 2019 and 2020 was more than five (Figure 2), which indicates that nitro-PAHs in the atmospheric PM_2.5_ in the summer in Guangzhou mainly come from the secondary formation of atmospheric photochemical reactions between parent PAHs and oxidants such as ·OH, NO_3_, and O_3_.

The spatial-temporal changes in the monomer ratios of the three nitro-PAHs in this study are displayed in Figure 3, which shows that 9-NT accounts for the highest proportion of nitro-PAHs. In the summer of 2019, the average proportions of nitro-PAHs in outdoor environments in Haizhu District were 98.0% (PM_2.5_) and 79.5% (TSP) and the average proportions in indoor environments were 94.3% and 72.2%. In Haizhu District, for PM_2.5_ and TSP, it was 78.21% and 89.09% (outdoor environment) and 69.51% and 80.68% (indoor environment), respectively (Figure 3). In contrast, 2-NF and 9-NT are the most abundant nitro-PAHs in urban and suburban atmospheres of the Mid-Atlantic region, accounting for approximately half of the total nitro-PAH concentration [41]. 9-Nitroanthracene is produced from direct sources such as diesel and gasoline exhausts and by the gas-phase reactions of PAHs with oxides of nitrogen [42]. 1-NP can be formed from the direct emission of diesel vehicles [31]. In addition, 9-NT can also be formed by the heterogeneous reaction of Ant produced in the combustion process (biomass combustion or vehicle emission) and a nitrating agent [42]. As a megacity in South China, Guangzhou has a large number of cars that can contribute large quantities of 9-NT. The high proportion of 9-NT during the study may come from the heterogeneous reaction of Ant and diesel vehicle emissions.

The previous studies reported that meteorological conditions also can influence the generation, accumulation, diffusion, removal, and phase partitioning of air pollutants in addition to emission sources from human activities (local and/or external) [43,44,45]. Appendix A summarizes the correlations between the meteorological conditions and the total mass concentration of nitro-PAHs in PM_2.5_ and TSP in the Haizhu District in the sampling periods. There are correlations between the total mass concentration of nitro-PAHs in PM_2.5_ and TSP in winter and temperature (R^2^ = 0.431–0.503, *p* < 0.05), solar radiation (R^2^ = 0.524–0.561, *p* < 0.05), and wind speed (R^2^ = 0.470–0.511, *p* < 0.05). Total nitro-PAH concentrations were significantly different with NO_2_ and O_3_ concentrations (*p* < 0.05) during both sampling periods, indicating that the heterogeneous photo-oxidation reactions of parent PAHs with atmospheric oxidants might be an important source. The low-latitude surface of Guangzhou receives high quantities of solar radiation, while the monsoon brings warm ocean air flow in summer. This forms a rainy climate with high temperatures and high humidity; in winter, the cold wind in the northern continent creates a dry and less rainy low-temperature climate. Comparing the correlation between nitro-PAHs and meteorological factors in summer and winter, nitro-PAHs in summer are more affected by the transfer and diffusion of PM_2.5_ and TSP. There may be a secondary generation reaction of nitro-PAHs in winter when the temperature and solar radiation are relatively high, which indicates that nitro-PAHs are correlated with meteorological parameters, unlike in summer. In addition, the secondary-generated nitro-PAHs in the atmosphere of the surrounding areas of Guangzhou (such as Dongguan, Shenzhen, Foshan, etc.) can also affect the concentration of nitro-PAHs in PM_2.5_ and TSP in the urban and suburban areas of Guangzhou through regional transfer. 

### 3.2. Health Risk Assessment

#### 3.2.1. Exposure Assessment

The ADD of three nitro-PAHs in different general populations (children and adults) in Guangzhou exposed to PM_2.5_ and TSP through the respiratory pathway was estimated according to Formula (3) and the calculation results of BaP_eq_ (Table 3). The ADDs of adult males, adult females, and children calculated based on the nitro-PAH concentrations in PM_2.5_ and TSP in Haizhu District of Guangzhou in the summer of 2019 were 5.81 × 10^−11^ ± 6.08 × 10^−12^ (PM_2.5_) and 3.65 × 10^−10^ ± 5.70 × 10^−11^ (TSP), 5.04 × 10^−11^ ± 5.27 × 10^−12^ and 3.17 × 10^−10^ ± 4.95 × 10^−11^, and 2.16 × 10^−11^ ± 2.26 × 10^−12^ and 1.35 × 10^−10^ ± 2.12 × 10^−11^ mg/(kg·d), respectively, while the ADDs in winter were 1.62 × 10^−10^ ± 2.26 × 10^−11^ (PM_2.5_) and 1.94 × 10^−9^ ± 5.72 × 10^−10^ (TSP), 1.40 × 10^−10^ ± 1.96 × 10^−11^ and 1.69 × 10^−9^ ± 4.96 × 10^−10^, and 5.99 × 10^−11^ ± 8.38 × 10^−12^ and 7.21 × 10^−10^ ± 2.12 × 10^−11^. According to the statistics, adult males receive the highest doses, followed by adult females, and children have the lowest doses. In the same period of study, the exposure doses of nitro-PAHs in PM_2.5_ and TSP in Huadu District were higher in summer than those in Haizhu District, while in winter the exposure doses of nitro-PAHs in PM_2.5_ were higher than those in Haizhu District (Table 3).

#### 3.2.2. Cancer Risk

The population is exposed to PM_2.5_- and TSP-containing nitro-PAHs through inhalation and the temporal and spatial changes in ILCR caused by respiratory exposure are shown in Table 4. The ILCRs of adult males, adult females, and children that were calculated based on the nitro-PAH concentrations in PM_2.5_ and TSP in Haizhu District of Guangzhou in the summer of 2019 were 1.83 × 10^−9^ ± 1.91 × 10^−11^ (PM_2.5_) and 1.15 × 10^−9^ ± 1.79 × 10^−10^ (TSP), 1.58 × 10^−10^ ± 1.66 × 10^−11^ and 9.95 × 10^−10^ ± 1.55 × 10^−10^, and 6.77 × 10^−11^ ± 7.08 × 10^−12^ and 4.25 × 10^−10^ ± 6.64 × 10^−11^, respectively, while the ILCRs in winter were 5.07 × 10^−10^ ± 7.09 × 10^−11^ (PM_2.5_) and 6.10 × 10^−9^ ± 1.80 × 10^−9^ (TSP), 4.40 × 10^−10^ ± 6.15 × 10^−11^ and 5.29 × 10^−9^ ± 1.56 × 10^−9^, and 1.88 × 10^−10^ ± 2.63 × 10^−11^ and 2.26 × 10^−9^ ± 6.66 × 10^−9^. According to the statistics, adult males have the highest ILCR, followed by adult females, and children have the lowest ILCR. In the same period of study, the ILCR induced by respiratory exposure to nitro-PAHs in PM_2.5_ and TSP in Huadu District was one order of magnitude higher than that in Haizhu District (Table 4).

## 4. Conclusions

Airborne particle phase concentrations of three nitro-PAHs were quantified in ambient air collected in the Haizhu District, an urban region, and Huadu District, a suburban area 33 km southeast of Haizhu District, from December 2019 to August 2020. The seasonal pattern of PM_2.5_- and TSP-bound nitro-PAHs in Guangzhou were predominantly ascribed to the changing of meteorological conditions. Nitro-PAHs showed no significant positive correlations with temporal variation from 2019 to 2020, indicating the influence of non-point source pollution. Concentrations at Huadu District were on average two to three times higher than those measured at the Haizhu District site. Concentrations for most nitro-PAHs were higher in summer than in winter, suggesting an increase in the secondary formation of various nitro-PAHs, which, when exposed to free radicals during summer, promoted the accumulation of nitro-PAHs. The relative contribution of gas-phase reactions and primary emission sources of nitro-PAHs were evaluated using source-specific concentration ratios of 2-NF/1-NP. The ratio of 2-NF/1-NP in TSP was less than five, while 2-NF/1-NP in outdoor PM_2.5_ in the summer of 2019 and 2020 was more than five, indicating that the secondary formation of atmospheric photochemical reactions between parent PAHs and oxidants were an important source of the three nitro-PAHs. 9-nitroanthracene was the most abundant of the nitro-PAHs quantitatively analyzed in the airborne particles at both sites, accounting for approximately 70–90% of the total nitro-PAH concentrations during summer and winter. The incremental lifetime cancer risks (ILCRs) of inhalation exposure to the three nitro-PAHs also indicated that the degree of health risk was still within the controllable range.

## Figures and Tables

**Figure 1 ijerph-19-15536-f001:**
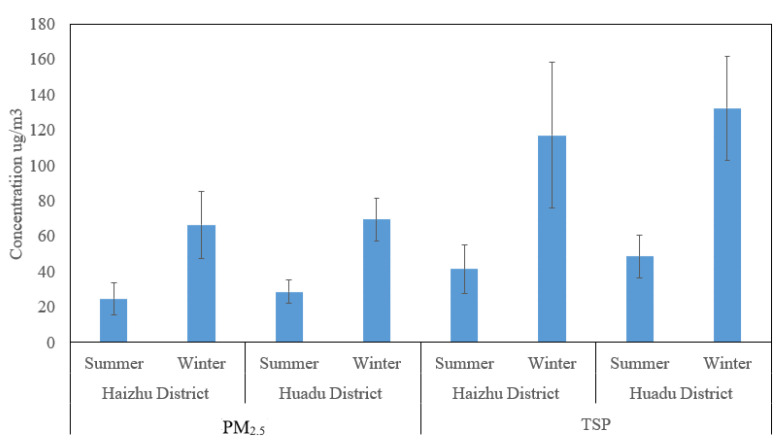
The mass concentrations of PM_2.5_ and TSP outdoors in 2019.

**Figure 2 ijerph-19-15536-f002:**
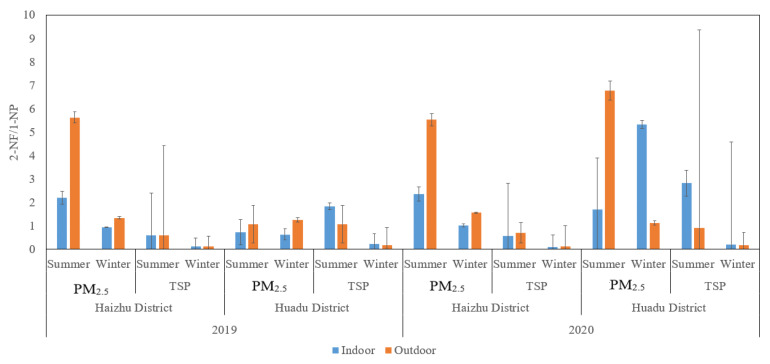
The diagnostic ratios of 2-NF/1-NP in PM_2.5_ and TSP during the sampling period.

**Figure 3 ijerph-19-15536-f003:**
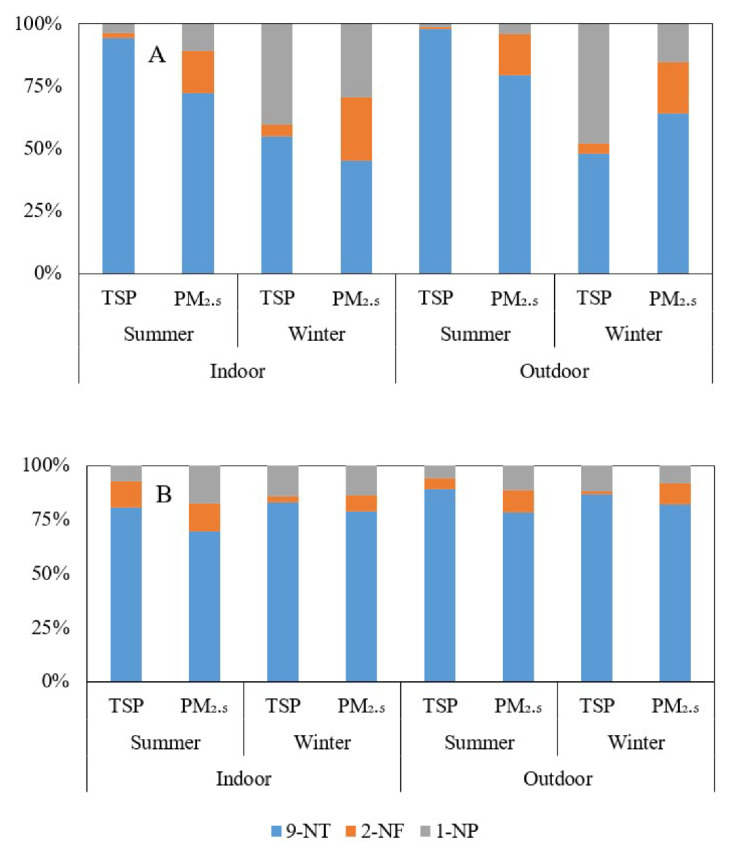
The composition of 9-NT, 2-NF, and 1-NP in PM_2.5_ and TSP in 2019 ((**A**): Haizhu District; (**B**): Huadu District).

**Table 1 ijerph-19-15536-t001:** Mean nitro-PAH concentrations (ng/m^3^) and standard deviations of 9-NT, 2-NF, and 1-NP in PM_2.5_ and TSP in 2019.

Nitro-PAHs	Summer				Winter			
	Indoor		Outdoor		Indoor		Outdoor	
	Mean	SD	Mean	SD	Mean	SD	Mean	SD
Haizhu District								
PM_2.5_								
9-NT ^a^	0.0612	0.0303	0.0918	0.0198	0.0683	0.0209	0.125	0.0203
2-NF ^b^	0.0144	0.00670	0.0189	0.0084	0.0388	0.0090	0.0402	0.0162
1-NP ^c^	0.0094	0.00410	0.00470	0.00320	0.0444	0.0104	0.0302	0.00650
TSP								
9-NT ^a^	0.734	0.144	1.09	0.243	0.548	0.105	0.645	0.191
2-NF ^b^	0.0151	0.00410	0.00870	0.00450	0.047	0.013	0.0564	0.0140
1-NP ^c^	0.0291	0.0101	0.0141	0.00170	0.400	0.109	0.640	0.252
Huadu District								
PM_2.5_								
9-NT ^a^	0.0894	0.0503	0.349	0.244	0.140	0.0293	0.192	0.0317
2-NF ^b^	0.0166	0.00680	0.046	0.0171	0.0130	0.00540	0.0236	0.0155
1-NP ^c^	0.0226	0.0067	0.0510	0.0178	0.0241	0.00990	0.0188	0.00410
TSP								
9-NT ^a^	0.518	0.223	0.795	0.139	0.384	0.0889	0.618	0.186
2-NF ^b^	0.0789	0.0223	0.0463	0.0171	0.0116	0.0109	0.0106	0.00290
1-NP ^c^	0.0451	0.0133	0.0510	0.0178	0.0667	0.0289	0.0837	0.0495

^a^, 9-NT: 9-nitroanthracene; ^b^, 2-NF: 2-nitrofluorene; ^c^, 1-NP: 1-nitropyrene.

**Table 2 ijerph-19-15536-t002:** Mean nitro-PAH concentrations (ng/m^3^) and standard deviations of 9-NT, 2-NF, and 1-NP in PM_2.5_ and TSP in 2020.

Nitro-PAHs	Summer				Winter			
	Indoor		Outdoor		Indoor		Outdoor	
	Mean	SD	Mean	SD	Mean	SD	Mean	SD
Haizhu District								
PM_2.5_								
9-NT ^a^	0.0845	0.0514	0.0928	0.0544	0.0861	0.0484	0.0933	0.0624
2-NF ^b^	0.0151	0.0092	0.0305	0.0156	0.0431	0.0232	0.0400	0.0316
1-NP ^c^	0.0122	0.0078	0.0056	0.0032	0.0497	0.0238	0.0239	0.0072
TSP								
9-NT ^a^	0.8462	0.3703	1.1768	0.7458	0.3720	0.2283	0.7865	0.5098
2-NF ^b^	0.0164	0.0023	0.0083	0.0042	0.0480	0.0357	0.0548	0.0197
1-NP ^c^	0.0351	0.0154	0.0132	0.0065	0.5713	0.6148	0.4495	0.1500
Huadu District								
PM_2.5_								
9-NT ^a^	0.0442	0.0290	0.1709	0.16844	0.0638	0.0301	0.0374	0.0194
2-NF ^b^	0.0842	0.0225	0.0274	0.0295	0.0170	0.0095	0.0289	0.0291
1-NP ^c^	0.0527	0.0161	0.0050	0.0016	0.0049	0.0031	0.0222	0.0187
TSP								
9-NT ^a^	0.6600	0.4440	0.8938	0.3440	0.5141	0.1653	0.9334	0.1686
2-NF ^b^	0.0819	0.02744	0.0432	0.0320	0.0141	0.0152	0.0145	0.0115
1-NP ^c^	0.0428	0.01211	0.0451	0.0233	0.0676	0.0259	0.0883	0.0578

^a^, 9-NT: 9-nitroanthracene; ^b^, 2-NF: 2-nitrofluorene; ^c^, 1-NP: 1-nitropyrene.

**Table 3 ijerph-19-15536-t003:** Average daily dose (ADD) of three nitro-PAHs (10^−10^) in Guangzhou from 2019 to 2020.

	District	PM_2.5_		TSP	
		Mean	SD	Mean	SD
2019					
Summer					
Males	Haizhu	0.581	0.0608	3.65	0.570
	Huadu	2.80	1.10	4.14	0.513
Females	Haizhu	0.504	0.0527	3.17	0.495
	Huadu	2.43	0.950	3.59	0.445
Children	Haizhu	0.216	0.0226	1.35	0.212
	Huadu	1.04	0.406	1.53	0.190
Winter					
Males	Haizhu	1.62	0.226	19.4	5.72
	Huadu	1.30	0.162	4.13	1.36
Females	Haizhu	1.40	0.196	16.9	4.96
	Huadu	1.13	0.140	3.58	1.18
Children	Haizhu	0.599	0.0838	7.21	2.12
	Huadu	0.484	0.0600	1.53	0.504
2020					
Summer					
Males	Haizhu	0.663	0.273	3.91	2.07
	Huadu	1.09	0.598	4.28	1.57
Females	Haizhu	0.575	0.236	3.39	1.79
	Huadu	0.944	0.519	3.71	1.36
Children	Haizhu	0.246	0.101	1.45	0.767
	Huadu	0.404	0.222	1.59	0.581
Winter					
Males	Haizhu	1.41	0.518	15.7	5.30
	Huadu	0.938	0.652	5.15	1.74
Females	Haizhu	1.22	0.449	13.6	4.59
	Huadu	0.813	0.566	4.47	1.50
Children	Haizhu	0.522	0.192	5.81	1.97
	Huadu	0.348	0.242	1.91	0.644

**Table 4 ijerph-19-15536-t004:** Incremental life cancer risk (ILCR) of three nitro-PAHs (10^−10^) in Guangzhou from 2019 to 2020.

	District	PM_2.5_		TSP	
		Mean	SD	Mean	SD
2019					
Summer					
Males	Haizhu	1.83	0.191	11.5	1.79
	Huadu	8.79	3.44	13.0	1.61
Females	Haizhu	1.58	0.166	9.95	1.55
	Huadu	7.62	2.98	11.3	1.40
Children	Haizhu	0.677	0.0708	4.25	0.664
	Huadu	3.26	1.28	4.82	0.598
Winter					
Males	Haizhu	5.07	0.709	61.0	18.0
	Huadu	4.09	0.508	13.0	4.27
Females	Haizhu	4.40	0.615	52.9	15.6
	Huadu	3.55	0.440	11.2	3.70
Children	Haizhu	1.88	0.263	22.6	6.66
	Huadu	1.52	0.188	4.81	1.58
2020					
Summer					
Males	Haizhu	2.08	0.856	12.3	6.49
	Huadu	3.42	1.88	13.4	4.92
Females	Haizhu	1.80	0.742	10.7	5.63
	Huadu	2.96	1.630	11.7	4.27
Children	Haizhu	0.772	0.317	4.56	2.41
	Huadu	1.27	0.697	4.99	1.83
Winter					
Males	Haizhu	4.42	1.63	49.2	16.6
	Huadu	2.94	2.05	16.2	5.45
Females	Haizhu	3.83	1.41	42.7	14.4
	Huadu	2.55	1.78	14.0	4.73
Children	Haizhu	1.64	0.604	18.2	6.17
	Huadu	1.09	0.760	6.00	2.02

## Data Availability

Data associated with the study are available upon request.

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
