# Peer review of "Health Risk Assessment of Inhalation Exposure to Airborne Particle-Bound Nitrated Polycyclic Aromatic Hydrocarbons in Urban and Suburban Areas of South China"

_ijerph, 2022, doi:10.3390/ijerph192315536_

Round 1

Reviewer 1 Report

The authors study the concentrations of three nitrated PAHs in the atmosphere of two parts of a Guangzhou city. While the differences between the two parts of the city and the sampling times did not seem to show much significance, I suspect that the more important part of this paper is to provide some kind of health risk assessment of these nitrated PAHs, which is important, given that these compounds are not studied all that much.

In general, the manuscript is well written and fairly easy to follow. The main issue however is going to be in the method. It is not clear how the replicates were performed or how PM2.5 and TMP were distinguished from each other. I think if these issues can be resolved, this should be suitable for publication.

In any case, please see my comments below.

Line 39: I think it should be made clear what the parent PAHs are reacting with here, i.e. NO3 radicals.

Lines 53-64: While this is nice detail for the introduction, having read the manuscript now, I think some of this information is superfluous. For example, nitronaphthalene doesn’t figure into your study, so the discussion on this could probably be taken out. I would just focus on the nitroPAHs that are pertinent to the study.

Lines 82-85: It seems that the distinction between PM2.5 and TSP is important for your study. However, it is not clear how you were able to separate these items. Given that one of your major findings is that nitroPAHs are more abundant in PM2.5 than in TSP, I think you should clarify how this is done.

Line 113: I apologize if I just missed it, but how many samples were actually taken for each outdoor site? It is not stated in section 2.1. I realize that not everyone has multiple high volume air samplers, so I think it should be made clear how many samples the averages (and associated error) in line 114 are based on. Were these samples taken in consecutive weeks? This information is  not made clear. Or are these averages based on analytical replicates?

Lines 164-165: I think this is a bit too strong. You really don’t have much evidence that wind speed and temperature is affecting the content of nitro-PAHs here.

Figure 2: As with my previous comment, I think the main issue to clarify so far is the sample counts. When I see these error bars, I have no idea how many samples these are based on. This seems to me the most egregious part of the manuscript.

Line 192: What do you mean here by the secondary reactions of diesel vehicles? Aren’t the emissions from diesel exhaust primary emissions?

Line 203: What do you mean by outdoor rings?

Author Response

Reviewer #1

Comments and Suggestions for Authors

Manuscript ijerph-1976304

The authors study the concentrations of three nitrated PAHs in the atmosphere of two parts of a Guangzhou city. While the differences between the two parts of the city and the sampling times did not seem to show much significance, I suspect that the more important part of this paper is to provide some kind of health risk assessment of these nitrated PAHs, which is important, given that these compounds are not studied all that much.

In general, the manuscript is well written and fairly easy to follow. The main issue however is going to be in the method. It is not clear how the replicates were performed or how PM2.5 and TMP were distinguished from each other.

Thank you for your suggestion. We have reflected this comment in line 107-112 by “A total of 208 samples and 16 field blanks in sampling points were simultaneously collected during winter (December 2019 and 2020) and summer (July to August 2019 and 2020). PM2.5 and TSP samples were respectively collected continuously using high-volume samplers (1000 L/min, Wuhan Tianhong Company, China) in outdoor areas and using medium-volume samplers (100 L/min, Qingdao Laoying Environment Technology Company, China) in indoor areas for 24 h.”.

I think if these issues can be resolved, this should be suitable for publication. In any case, please see my comments below.

Line 39: I think it should be made clear what the parent PAHs are reacting with here, i.e. NO3 radicals.

We have reflected this comment in line 59-67 by “Several studies have reported another formation pattern of nitro-PAHs, which was through gas phase reactions between parent-PAHs and oxidants radicals, such as OH, NO3and O3 [8-10]. In the atmosphere, it is widely assumed that OH or NO3 radicals would attack the C atoms on the aromatic rings in the PAH molecules followed by addition of NO2 to form the OH-PAH or NO3-PAH adducts at the ortho position, and the loss of water or nitric acid to nitro-PAHs. The OH-PAH-NO2 will lose H2O via four-membered transition state to form nitro-PAH, and the NO3-PAH-NO2 adduct will lose HNO3 via six-membered transition state to form nitro-PAH [9-12].”

[8] Zimmermann, K., Jariyasopit, N., Simonich, S.L.M., Tao, S., Atkinson, R., Arey, J., 2013.Formation of nitro-PAHs from the heterogeneous reaction of ambient particle-bound PAHs with N2O5/NO3/NO2. Environ. Sci. Technol. 2013. 47, 8434-8442.

[12] Zhang Qingzhu, Gao Rui, Xu Fei, Zhou Qin, Jiang Guibin, Wang Tao, Chen Jianmin, Hu Jingtian, Wang Wenxing. Role of water molecule in the gas-phase formation process of nitrated polycyclic aromatic hydrocarbons in the atmosphere: A computational study. Environ. Sci. Technol.2014, 48, (9), 5051-5057.

Lines 53-64: While this is nice detail for the introduction, having read the manuscript now, I think some of this information is superfluous. For example, nitronaphthalene doesn’t figure into your study, so the discussion on this could probably be taken out. I would just focus on the nitroPAHs that are pertinent to the study.

We agree with you and have deleted this sentence in the manuscript.

Lines 82-85: It seems that the distinction between PM2.5 and TSP is important for your study. However, it is not clear how you were able to separate these items. Given that one of your major findings is that nitroPAHs are more abundant in PM2.5 than in TSP, I think you should clarify how this is done.

Thank you for your suggestion. We have corrected this error in line 109-112 by “PM2.5 and TSP samples were respectively collected continuously using high-volume samplers (1000 L/min, Wuhan Tianhong Company, China) in outdoor areas and using medium-volume samplers (100 L/min, Qingdao Laoying Environment Technology Company, China) in indoor areas for 24 h.”.

Line 113: I apologize if I just missed it, but how many samples were actually taken for each outdoor site? It is not stated in section 2.1. I realize that not everyone has multiple high volume air samplers, so I think it should be made clear how many samples the averages (and associated error) in line 114 are based on. Were these samples taken in consecutive weeks? This information is not made clear. Or are these averages based on analytical replicates?

Thank you for your suggestion. We have supplied the content in line 107-109 by “A total of 208 samples and 16 field blanks in sampling points were simultaneously collected during winter (December 2019 and 2020) and summer (July to August 2019 and 2020).”

Lines 164-165: I think this is a bit too strong. You really don’t have much evidence that wind speed and temperature is affecting the content of nitro-PAHs here.

We agree with you. We have deleted these sentences in the manuscript.

Figure 2: As with my previous comment, I think the main issue to clarify so far is the sample counts. When I see these error bars, I have no idea how many samples these are based on. This seems to me the most egregious part of the manuscript.

We agree with your assessment. We have reflected this comment in line 107-112 by “A total of 208 samples and 16 field blanks in sampling points were simultaneously collected during winter (December 2019 and 2020) and summer (July to August 2019 and 2020). PM2.5 and TSP samples were respectively collected continuously using high-volume samplers (1000 L/min, Wuhan Tianhong Company, China) in outdoor areas and using medium-volume samplers (100 L/min, Qingdao Laoying Environment Technology Company, China) in indoor areas for 24 h.”.

Line 192: What do you mean here by the secondary reactions of diesel vehicles? Aren’t the emissions from diesel exhaust primary emissions?

We have incorporated your comments in line 262-265 by “9-Nitroanthracene is produced from direct sources such as diesel, gasoline exhaust and by the gas-phase reactions of PAHs with oxides of nitrogen [42].1-NP can be formed from the direct emission of diesel vehicles [31].”

Line 203: What do you mean by outdoor rings?

We have revised the text (lines 272-294) to reflect the relationship between PM2.5-/TSP-bound nitro-PAHs with environmental monitoring parameters by “The previous studies reported that meteorological conditions also can influence the generation, accumulation, diffusion, removal, and phase partitioning of air pollutants in addition to emission sources from human activities (local and/or external) [43-45]. Table S3 summarizes the correlations between the meteorological conditions and the total mass concentration of nitro-PAHs in PM2.5 and TSP in the Haizhu District in the sampling periods. There are correlations between the total mass concentration of nitro-PAHs in PM2.5 and TSP in winter and temperature (R2=0.43-0.50, p < 0.05), solar radiation (R2=0.52-0.56, p < 0.05) and wind speed (R2=0.47-0.51, p < 0.05). Total nitro-PAHs concentrations were significantly different with NO2 and O3 concentrations (p < 0.05) during the both sampling period indicating that heterogeneous photo-oxidation reactions of parent PAHs with atmospheric oxidant might be an important source. The low-latitude surface of Guangzhou receives high quantities of solar radiation, while the monsoon brings warm ocean air flow in summer. This forms a rainy climate with high temperatures and high humidity; in winter, the cold wind in the northern continent creates a dry and less rainy low-temperature climate. Comparing the correlation between nitro-PAHs and meteorological factors in summer and winter, nitro-PAHs in summer are more affected by the transfer and diffusion of PM2.5 and TSP. There may be a secondary generation reaction of nitro-PAHs in winter when the temperature and solar radiation are relatively high, which indicates that nitro-PAHs are correlated with meteorological parameters, unlike in summer. In addition, the secondary-generated nitro-PAHs in the atmosphere of the surrounding areas of Guangzhou (such as Dongguan, Shenzhen, Foshan, etc.) can also affect the concentration of nitro-PAHs in PM2.5 and TSP in the urban and suburban areas of Guangzhou through regional transfer.”.

Reviewer 2 Report

The article entitled “Health Risk Assessment of Inhalation Exposure to Airborne Particle–bound Nitrated Polycyclic Aromatic Hydrocarbons in Urban and Suburban Areas of South China” studied the behaviour of PM2.5 and TPS outdoors and indoors in urban and suburban areas of southern China, and studied the concentration of nitro-PAH in samples of PM2.5 and TPS. Finally, in this study, the Health Risk Assessment was calculated through the total equivalent carcinogenic concentration and the average daily exposure of toxic substances in PM2.5 and TSP.

The results obtained by this study showed that there is no significant difference between PAH concentrations in summer and winter, and that the incremental lifetime cancer risk is acceptable, according to the lower limit established by the US EPA. This article highlights the importance of monitoring nitro-PAHs in areas where air emissions from combustion processes are significant throughout the year. This type of study is relevant to understand the health risk that can be caused by air pollution and in particular by nitro-PAHs, and the calculations, although theoretical, are an approach to the behaviour and impact of these particles in the respiratory system of the human being.

Line 66: the unit pg/m3

Line 164: You mentioned that “This study showed that the content of nitro-PAHs is high in cold weather and is greatly affected by temperature and wind speed”, however this relationship between nitro-PAHs and temperature and wind speed is not showed in the article. Can you provide any table or figure that prove this?

Can you add a brief explanation about the statistical methodology and the identification methodology of PAHs used?

Conclusions can be improved: The conclusions could be improved. They must reflect the general conclusions of the work and not a summary of the results obtained.

Author Response

Reviewer #2

Comments and Suggestions for Authors

The article entitled “Health Risk Assessment of Inhalation Exposure to Airborne Particle–bound Nitrated Polycyclic Aromatic Hydrocarbons in Urban and Suburban Areas of South China” studied the behaviour of PM2.5 and TPS outdoors and indoors in urban and suburban areas of southern China, and studied the concentration of nitro-PAH in samples of PM2.5 and TPS. Finally, in this study, the Health Risk Assessment was calculated through the total equivalent carcinogenic concentration and the average daily exposure of toxic substances in PM2.5 and TSP. The results obtained by this study showed that there is no significant difference between PAH concentrations in summer and winter, and that the incremental lifetime cancer risk is acceptable, according to the lower limit established by the US EPA. This article highlights the importance of monitoring nitro-PAHs in areas where air emissions from combustion processes are significant throughout the year. This type of study is relevant to understand the health risk that can be caused by air pollution and in particular by nitro-PAHs, and the calculations, although theoretical, are an approach to the behaviour and impact of these particles in the respiratory system of the human being.

Line 66: the unit pg/m3

We have corrected this error in the manuscript.

Line 164: You mentioned that “This study showed that the content of nitro-PAHs is high in cold weather and is greatly affected by temperature and wind speed”, however this relationship between nitro-PAHs and temperature and wind speed is not showed in the article. Can you provide any table or figure that prove this?

We removed [This study showed that the content of nitro-PAHs is high in cold weather and is greatly affected by temperature and wind speed, the low ring number (<3) nitro-PAHs in the at-mosphere are mainly concentrated in the gaseous phase, and the multiple ring number (>3) is concentrated in the particulate phase. 2-NF and 9-NT are the most abundant nitro-PAHs in the atmosphere, accounting for approximately half of the total nitro-PAHs concentration. Light is important to the decomposition of nitro-PAHs.] and hope that the deletion clarifies the points we attempted to make. In this section, based on diagnostic ratios and specific molecular markers, sources of nitro-PAHs with different airborne particle levels in Guangzhou were identified.

Can you add a brief explanation about the statistical methodology and the identification methodology of PAHs used?

We agree with you and have supplied the content in line 197-199 by

“2.6. Statistical analysis

The PM2.5- and TSP-bound nitro-PAHs concentrations and profiles were compared using IBM SPSS Statistics 20.0 software, with a statistical significance of p < 0.05.”

Conclusions can be improved: The conclusions could be improved. They must reflect the general conclusions of the work and not a summary of the results obtained.

Thank you for providing this insight. We have reflected this comment in line 328-348 by “Airborne particle phase concentrations of 3 nitro-PAHs were quantified in ambient air collected in Haizhu District, an urban region, and in Huadu District, a suburban area 33 km southeast of Haizhu District, from December 2019 to August 2020. The seasonal pattern of PM2.5- and TSP bound nitro-PAHs in the Guangzhou were predominantly ascribed to the changing of meteorological conditions. Nitro-PAHs showed no significantly positive correlations with temporal variation from 2019 to 2020, indicating the influence of non-point source pollution. Concentrations at Huadu District were on average two to three times higher than those measured at the Haizhu District site. Concentrations for most nitro-PAHs were higher in summer than in winter, suggesting a increase the secondary formation of various nitro-PAHs when exposed to free radicals during summer promoted the accumulation of nitro-PAHs. The relative contribution of gas-phase reactions and primary emission sources of nitro-PAHs were evaluated using source specific concentration ratios of 2-NF/1-NP. 2-NF/1-NP in TSP is less than 5, while 2-NF/1-NP in outdoor PM2.5 in summer of 2019 and 2020 is more than 5, indicating the secondary formation of atmospheric photochemical reactions between parent PAHs and oxidants were an important source of 3 nitro-PAHs. 9-nitroanthracene was the most abundant of the nitro-PAHs quantitatively analyzed in the airborne particle at both sites, accounting for approximately 70%-90% of the total nitro-PAH concentrations during summer and winter. The Incremental Lifetime Cancer Risks (ILCRs) of inhalation exposure to 3 nitro-PAHs also indicated, the degree of health risk was still within the controllable range.”

Reviewer 3 Report

Manuscript ijerph-1976304

Journal: International Journal of Environmental Research and Public Health Manuscript ID: ijerph-1976304 Type of manuscript: Article; Submitted to section: Environmental Science and Engineering.

Title: Health Risk Assessment of Inhalation Exposure to Airborne Particle–bound Nitrated Polycyclic Aromatic Hydrocarbons in Urban and Suburban Areas of South China.

Authors: Peng Gao, Feng Deng, Weishan Chen, Yijia Zhong, Xiaolu Cai, Wenmin Ma, Jian Hu, Shuran Feng.

This study evaluates exposure to different size PM-bound Nitrated PAHs in South China during one year.  For that purpose, the authors characterize selected three Nitrated PAH levels in indoor and outdoor conditions in sites of various types, evaluate long-range transport potential of these species through their mass size distribution, and estimate their inhalation bio-accessibility to assess the health risk of nitro-PAH pollution in the studied sites.  Although the study is scientifically sound, it misses lots of important components to support the findings and discussion, for instance, statistical data, the names and values of the statistical tests (correlation) etc.  Before accepting for publishing, I recommend a major revision of this manuscript. 

Minor comments:

Lines 42-43:  “. Benzo[a]pyrene (BaP) and 1-nitropyrene (1-NP) are categorized into Groups 1….” Why you mention BaP here, it is out of the context.

Line 62:  “.some studies….”, consider: Some studies….

Line 66:  “several p/g m3 to…..”, consider: pg/m3.

Line 94:  “a previous study [17,25,26,30,31].”, consider “in previous studies [17,25,26,30,31]”.

Line 105:  The average recoveries were 87-118% for NPAHs.”, please consider “The recoveries were 87-123 % for NPAHs”.

Lines 113-117:  “….the average concentration of PM2.5 outdoors in Haizhu district of Guangzhou was 24.7 ± 9.12 and 66.5 ± 18.8 μg/m3, respectively, and in Huadu District was 28.6 ± 6.60 and 69.6 ± 12.0 μg/m3; the mass concentration of PM2.5 of Huadu District in winter exceeded the PM2.5 the state air quality standard (GB 30952012) limit of 35μg/m3 in (Figure 1).”, unclear statement:  Figure 1 shows that 35μg/m3  objective concentration was exceeded in both locations in winter, not only in Huadu.

Line 126:  “…shown in Table 1 and 2.”, consider “shown in Tables 1 and 2.”

Lines 151-153 Table:  consider correcting the numbers in the table and text below for significant digits.

Lines 129-131:  “In the winter of 2019, the average nitro-PAH concentrations in PM2.5 were 0.195 ± 0.0176 and 0.152 ± 0.0230 ng/m3, respectively, and nitro-PAHs in TSP were 1.34 ± 0.202 and 0.996 ± 0.0643 ng/m3, respectively.”, Respectively to what? consider adding: outdoors and indoors.

Line 133 and 159:  “…Guangzhou, but the differences were not statistically significant.”, Consider adding:  “…significant (test name, p < value).

Lines 167-168:  “2-NF and 9-NT are the most abundant nitro-PAHs in the atmosphere, accounting for approximately half of the total nitro-PAHs concentration”, Cite references to support your statement.

Line 181:  “….than 5 (Figure 5),”, consider “(Figure 2)”.

 Line 194:  “correlation between 9-NT and 1-NP”, consider showing the correlation test and correlation value.

Lines 212-217:  “…there are correlations between the total mass concentration of nitro-PAHs in PM2.5 and TSP in winter and temperature, solar radiation and wind speed (p < 0.05), while there are no correlations between the total mass concentration of nitro-PAHs in PM2.5 and TSP in summer and meteorological factors”,

(p < 0.05) seems to be a wrong value, in addition sometimes R2 is used for correlations.  Consider providing the correlation test name and correlation value here.

Lines 212-227:  It would be supportive to see a table in SI with correlations between N-PAH concentrations and Temp, solar radiation, and wind speed in summer and winter to support your statements in the Discussion, otherwise this discussions is useless.

Lines 240-263:  consider moving this theoretical part to the Introduction Section, Discussion Section is for the results discussion.

Lines 310-321:  the Conclusions are poor, the authors repeated their findings here, but what are the conclusions from the findings?

 Supplementary Information:

1. 2.Materials and Methods

“The injected volume was 1 mL in splitless mode.”, consider 1µL.

2. “3 nitro-PAHs were detected using an Agilent 7000A GC/MS Triple Quadrupole System (Agilent Technologies Inc, USA) and quantitatively analyzed in MRM mode. Ion”, Please provide quantifying and qualifying ions, and acceptance criteria.

3. Please justify why recovery standards (Nap-d8; Phe-d10) were analyzed in different runs and on different instruments than the original air samples.  Please indicate if you corrected the results for recoveries of surrogates.

Author Response

Reviewer #3

Comments and Suggestions for Authors

Manuscript ijerph-1976304

Journal: International Journal of Environmental Research and Public Health Manuscript ID: ijerph-1976304 Type of manuscript: Article; Submitted to section: Environmental Science and Engineering.

Title: Health Risk Assessment of Inhalation Exposure to Airborne Particle–bound Nitrated Polycyclic Aromatic Hydrocarbons in Urban and Suburban Areas of South China.

Authors: Peng Gao, Feng Deng, Weishan Chen, Yijia Zhong, Xiaolu Cai, Wenmin Ma, Jian Hu, Shuran Feng.

This study evaluates exposure to different size PM-bound Nitrated PAHs in South China during one year. For that purpose, the authors characterize selected three Nitrated PAH levels in indoor and outdoor conditions in sites of various types, evaluate long-range transport potential of these species through their mass size distribution, and estimate their inhalation bio-accessibility to assess the health risk of nitro-PAH pollution in the studied sites. 

Although the study is scientifically sound, it misses lots of important components to support the findings and discussion, for instance, statistical data, the names and values of the statistical tests (correlation) etc. 

Thank you for providing these insights. We agree with you and have incorporated this suggestion throughout our paper.

“2.6. Statistical analysis

The PM2.5- and TSP-bound nitro-PAHs concentrations and profiles were compared using IBM SPSS Statistics 20.0 software, with a statistical significance of p < 0.05.”

Before accepting for publishing, I recommend a major revision of this manuscript.

Minor comments:

Lines 42-43: “. Benzo[a]pyrene (BaP) and 1-nitropyrene (1-NP) are categorized into Groups 1….” Why you mention BaP here, it is out of the context.

We have reflected this comment in line 70-71 by “1-nitropyrene (1-NP) is categorized into Groups 2A (probably carcinogenic to humans) [9].”

Line 62: “.some studies….”, consider: Some studies….

We have corrected this error by “ Some studies reported that the concentrations of nitro-PAHs emitted from diesel engines are much higher than those from gasoline engines [25,26].”

Line 66: “several p/g m3 to…..”, consider: pg/m3.

We have corrected this error in the manuscript.

Line 94: “a previous study [17,25,26,30,31].”, consider “in previous studies [17,25,26,30,31]”.

We have incorporated your comments in line 123 by “The pretreatment of filters is similar to that in previous study [32-34].”

Line 105: “The average recoveries were 87-118% for NPAHs.”, please consider “The recoveries were 87-123 % for NPAHs”.

We have reflected this comment in line 134 by “The recoveries were 87-118 % for nitro-PAHs.”

Lines 113-117:  “….the average concentration of PM2.5 outdoors in Haizhu district of Guangzhou was 24.7 ± 9.12 and 66.5 ± 18.8 μg/m3, respectively, and in Huadu District was 28.6 ± 6.60 and 69.6 ± 12.0 μg/m3; the mass concentration of PM2.5 of Huadu District in winter exceeded the PM2.5 the state air quality standard (GB 3095-2012) limit of 35μg/m3 in (Figure 1).”, unclear statement: Figure 1 shows that 35μg/m3 objective concentration was exceeded in both locations in winter, not only in Huadu.

Thank you for your suggestion. We have reflected this comment in line 201-205 by “In the summer and winter seasons of 2019, the average concentration of PM2.5 outdoors in Haizhu district of Guangzhou was 24.7 ± 9.12 and 66.5 ± 18.8 μg/m3, respectively, and in Huadu District was 28.6 ± 6.60 and 69.6 ± 12.0 μg/m3; Figure 1 shows that 35μg/m3 objective concentration (GB 3095-2012) was exceeded in both locations in winter.”

Line 126:  “…shown in Table 1 and 2.”, consider “shown in Tables 1 and 2.”

We have corrected this error by “The temporal and spatial variations in nitro-PAH concentrations in PM2.5 and TSP in Guangzhou are shown in Tables 1 and 2.”

Lines 151-153 Table: consider correcting the numbers in the table and text below for significant digits.

We have incorporated the three significant digits throughout our paper.

Lines 129-131: “In the winter of 2019, the average nitro-PAH concentrations in PM2.5 were 0.195 ± 0.0176 and 0.152 ± 0.0230 ng/m3, respectively, and nitro-PAHs in TSP were 1.34 ± 0.202 and 0.996 ± 0.0643 ng/m3, respectively.”, Respectively to what? consider adding: outdoors and indoors.

Thank you for your suggestion. We have reflected this comment in line 215-217 by “In the winter of 2019, the average nitro-PAH concentrations in PM2.5 in Haizhu District were 0.195 ± 0.0176 (outdoors) and 0.152 ± 0.0230 (indoors) ng/m3, and nitro-PAHs in TSP were 1.34 ± 0.202 and 0.996 ± 0.0643 ng/m3.”

Line 133 and 159: “…Guangzhou, but the differences were not statistically significant.”, Consider adding: “…significant (test name, p < value).

We agree with you and have incorporated this suggestion in line 223-225 by “There were differences between the average concentrations of nitro-PAHs in PM2.5 and TSP from different districts of Guangzhou, but the differences were not statistically significant (p>0.05).”

Lines 167-168: “2-NF and 9-NT are the most abundant nitro-PAHs in the atmosphere, accounting for approximately half of the total nitro-PAHs concentration”, Cite references to support your statement.

We have removed [In contrast, 2-NF and 9-NT are the most abundant nitro-PAHs in urban and suburban atmospheres of the Mid-Atlantic region, accounting for approximately half of the total nitro-PAHs concentration [40]] (lines 260-262) to be more in line with your comments. We hope that the edited section clarifies the composition of 3 nitro-PAHs.

[40] Bamford, Holly A.; Baker, Joel E. Nitro-polycyclic aromatic hydrocarbon concentrations and sources in urban and suburban atmospheres of the Mid-Atlantic region. Atmos. Environ. 2003, 37, 2077-2091.

Line 181: “….than 5 (Figure 5),”, consider “(Figure 2)”.

We have corrected this error in the manuscript.

Line 194: “correlation between 9-NT and 1-NP”, consider showing the correlation test and correlation value.

Thank you for your suggestion. We have deleted this sentence.

Lines 212-217: “…there are correlations between the total mass concentration of nitro-PAHs in PM2.5 and TSP in winter and temperature, solar radiation and wind speed (p < 0.05), while there are no correlations between the total mass concentration of nitro-PAHs in PM2.5 and TSP in summer and meteorological factors”, (p < 0.05) seems to be a wrong value, in addition sometimes R2 is used for correlations. Consider providing the correlation test name and correlation value here.

We have incorporated your comments in line 271-281 by “The previous studies reported that meteorological conditions also can influence the generation, accumulation, diffusion, removal, and phase partitioning of air pollutants in addition to emission sources from human activities (local and/or external) [43-45]. Table S3 summarizes the correlations between the meteorological conditions and the total mass concentration of nitro-PAHs in PM2.5 and TSP in the Haizhu District in the sampling periods. There are correlations between the total mass concentration of nitro-PAHs in PM2.5 and TSP in winter and temperature (R2=0.431-0.503, p < 0.05), solar radiation (R2=0.524-0.561, p < 0.05) and wind speed (R2=0.470-0.511, p < 0.05). Total nitro-PAHs concentrations were significantly different with NO2 and O3 concentrations (p < 0.05) during the both sampling period indicating that heterogeneous photo-oxidation reactions of parent PAHs with atmospheric oxidant might be an important source.”

[42] Amarillo A.C.; Carreras H. Quantifying the influence of meteorological variables on particle-bound PAHs in urban environments. Atmos. Pollut. Res. 2016,7, 597–602.

[43] Wang H.L.; Qiao L.P.; Lou S.R.; Zhou M.; Ding A.J.; Huang H.Y.; Chen J.M.; Wang Q.; Tao S.K.; Chen C.H.; Li L.; Huang C. Chemical composition of PM2.5 and meteorological impact among three years in urban Shanghai, China. J. Clean. Product. 2016, 112, 1302-1311.

[44] Hong Y.W.; Xu X.B.; Liao D.; Ji X.T.; Hong Z.Y.; Chen Y.T.; Xu L.L.; Li M.R.; Wang H.; Zhang H.; Xiao H.; Choi S.D.; Chen J.S. Air pollution increases human health risks of PM2.5-bound PAHs and nitro-PAHs in the Yangtze River Delta, China. Sci. Total Environ. 2021,770,145402.

Lines 212-227: It would be supportive to see a table in SI with correlations between N-PAH concentrations and Temp, solar radiation, and wind speed in summer and winter to support your statements in the Discussion, otherwise this discussions is useless.

We agree with you and have supplied the Table S3 in the Supplementary Information.

Table S3 Correlation coefficients and p-values between total nitro-PAHs and meteorological parameters with different airborne particle in Haizhu District in 2019.

Nitro-PAHs

SO2

CO

NO2

O3

Temperature

Solar radiation

Wind speed

Winter in 2019

PM2.5

0.171

0.302

0.404*

0.441*

0.431*

0.561*

0.511*

TSP

0.262

0.263

0.471*

0.383*

0.503*

0.524*

0.470*

Summer in 2019

PM2.5

-0.344

0.21

0.450*

0.531*

-0.270

0.550*

-0.442*

TSP

0.224

0.33

0.502*

0.410*

-0.310

0.492*

-0.381

∗ means correlation is significant at p < 0.05

Lines 240-263: consider moving this theoretical part to the Introduction Section, Discussion Section is for the results discussion.

Thank you for providing these insights. We have reflected this comment in line 140-195 by “2.4. Exposure assessment

The respiratory pathway is the main method by which nitro-PAHs enter the human body. The various components of nitro-PAHs have different toxicities, with differing risks of harm to the human respiratory system. Benzo[a]pyrene (BaP) has been identified as a strong carcinogen. Previous studies measured a more reasonable and accurate ratio of the carcinogenic toxicity of nitro-PAHs to BaP through toxicological tests and proposed the concept of toxic equivalence factors (TEFs) [35,36]. This study referred to the relative study results of TEFs of nitro-PAHs and calculated the total equivalent carcinogenic concentration (BaP equivalent concentration, BaPeq) of 3 kinds of nitro-PAHs with BaP as the reference. This aimed to assess the health risk of nitro-PAH pollution in PM2.5 and TSP. The calculation method is shown in Formula (1).

BaPeq =  Ci x TEFi (1)

BaPEQ is the total equivalent carcinogenic concentration with BaP as the reference, ng/m3; C is the concentration of the i-th nitro-PAHs, ng/m3; TEF is the carcinogenic equivalent factor of the i-th nitro-PAHs. Carcinogenic and mutagenic equivalent factors of nitro-PAHs are shown in Table S1.

The Average Daily Dose (ADD) refers to the daily intake of nitro-PAHs via the respiratory pathway. This study mainly considers the exposure dose of nitro-PAHs through respiratory exposure and the lifetime cancer risk of the population due to nitro-PAHs. Due to differences in ages, sexes, physiological characteristics and lifestyle habits, the population was divided into children, adult males and adult females for this study. The ADD of respiratory exposure pathways by different people can be calculated by Formula (2):

ADD = (C x IR x F x EF x ED)/(BW x AT)   (2)

ADD is the average daily exposure of toxic substances in PM2.5 and TSP mg/(kg·d); C is the concentration of toxic substances (e.g., nitro-PAHs) in PM2.5 and TSP. This study is based on the BaP total equivalent carcinogenic concentration of the bioavailable concentration released by toxic substances in simulated human lung fluid (ng/m3); IR is respiratory volume m3/d; EF is the exposure frequency (day/year); ED is exposure duration (year); F is the indoor and outdoor stay time; BW is body weight (kg); and AT is the average exposure time (day). The exposure parameter is the key factor in determining the accuracy of environmental health risk assessment. The closer the selected exposure parameter is to the actual situation of the target population, the more authentic the results of environmental health risk assessment will be. According to the manual of Chinese population exposure parameters issued by the Ministry of Environmental Protection of China at the end of 2013, and combined with the actual situation of Guangzhou, the population exposure evaluation parameters are determined. The specific parameters are shown in Table S2.

2.5. Health risk assessment

  According to the US EPA’s health risk assessment model for respiratory exposure to carcinogenic and toxic substances, the incremental life cancer risk (ILCR) of nitro-PAHs in PM2.5 and TSP [20,37] is as follows:

ILCR= q x ADD (3)

q is the human carcinogenic intensity coefficient calculated according to animal experimental data, kg·d/mg; ADD is Average Daily Dose, mg/(kg·d). According to the US EPA IRIS, the carcinogenic intensity coefficient (q) of BaP through respiratory exposure is 3.14 kg·d/mg. The US EPA proposes that the level of Incremental Lifetime Cancer Risk (ILCR) of pollutants is acceptable between 10-4-10-6; that is, the exposure level is acceptable when the number of health hazards or deaths caused by exposure to pollutants is not more than 1 per 10000 to 1 million people. An ILCR lower than 10-6 indicates that the risk degree is not significant, and generally the necessity of risk management is not great; an ILCR of 10-6-10-5 indicates that a risk is higher than the risk of daily activities and deserves attention; an ILCR of 10-5-10-4 indicates that close attention needs to be paid and a solution should be adopted; and an ILCR higher than 10-4 indicates that the risk degree is significant, which requires the adoption risk management to treat.”.

Lines 310-321: the Conclusions are poor, the authors repeated their findings here, but what are the conclusions from the findings?

 We agree with you. We have reflected this comment in line 327-347 by “Airborne particle phase concentrations of 3 nitro-PAHs were quantified in ambient air collected in Haizhu District, an urban region, and in Huadu District, a suburban area 33 km southeast of Haizhu District, from December 2019 to August 2020. The seasonal pattern of PM2.5- and TSP bound nitro-PAHs in the Guangzhou were predominantly ascribed to the changing of meteorological conditions. Nitro-PAHs showed no significantly positive correlations with temporal variation from 2019 to 2020, indicating the influence of non-point source pollution. Concentrations at Huadu District were on average two to three times higher than those measured at the Haizhu District site. Concentrations for most nitro-PAHs were higher in summer than in winter, suggesting a increase the secondary formation of various nitro-PAHs when exposed to free radicals during summer promoted the accumulation of nitro-PAHs. The relative contribution of gas-phase reactions and primary emission sources of nitro-PAHs were evaluated using source specific concentration ratios of 2-NF/1-NP. 2-NF/1-NP in TSP is less than 5, while 2-NF/1-NP in outdoor PM2.5 in summer of 2019 and 2020 is more than 5, indicating the secondary formation of atmospheric photochemical reactions between parent PAHs and oxidants were an important source of 3 nitro-PAHs. 9-nitroanthracene was the most abundant of the nitro-PAHs quantitatively analyzed in the airborne particle at both sites, accounting for approximately 70%-90% of the total nitro-PAH concentrations during summer and winter. The Incremental Lifetime Cancer Risks (ILCRs) of inhalation exposure to 3 nitro-PAHs also indicated, the degree of health risk was still within the controllable range.”

Supplementary Information:

  1. 2. Materials and Methods

“The injected volume was 1 mL in splitless mode.”, consider 1µL.

We have reflected this comment by “The injected volume was 1 µL in splitless mode.”.

  1. “3 nitro-PAHs were detected using an Agilent 7000A GC/MS Triple Quadrupole System (Agilent Technologies Inc, USA) and quantitatively analyzed in MRM mode. Ion”, Please provide quantifying and qualifying ions, and acceptance criteria.

We agree with you and have incorporated this suggestion throughout our Supplementary information by “ 2.2. Sample analysis

The analysis of nitro-PAHs and PAHs required two instrumental runs. Total 3 nitro-PAHs were detected using an Agilent 7000A GC/MS Triple Quadrupole System (Agilent Technologies Inc, USA) and quantitatively analyzed in MRM mode. Ion source: inert ion source (EI). Chromatographic column: Agilent 19091J-433 HP-5 (30 m x 0.25mm x 0.25 mm). The injected volume was 1 µL in splitless mode. The parameters were as follows: inlet temperatures were 240 ◦C for nitro-PAHs, 80 oC ramp to 240 oC (at a rate of 20 oC /min) and held for 7.5 min, then ramp to 300 oC (at a rate of 30 oC /min) and held for 2.5 min; the initial temperature of GC was 80 ◦C (held 3 min) for PAHs (Naphthalene-d8 and phenanthrene-d10), and then heat to 200 oC (at a rate of 10 oC /min) and held for 4 min, last heat to 300 oC (at a rate of 6 oC /min) and held for 8 min. Due to various isomers were contained in detected nitro-PAHs and PAHs, different oven temperature program were tried to select the optimal method, which has not only the better analytical peaks but also the shorter separation time [1]. The initial collision energy (CE) was set as 10–60 eV to determine the characteristic fragment ions. Lastly, CE was further optimized with the increase in sensitivity under Multi Reaction Monitor (MRM) mode. Different CE parameters were selected from 5 eV to 60 e V and the other parameters were same, appropriate CE parameter was determined by observing the corresponding signal conditions of each target peak for 3 nitro-PAHs [1].

3 nitro-PAHs were separated and the analysis time was 26.397 min for 2-Nitrofluorene (CE:15 eV), 27.170 min for 9-Nitroanthracene (CE:15 eV) and 32.241 min for 1-Nitropyrene (CE:25 eV).”

[1] Zhang Z.F.; Chen J.C.; Zhao Y.X.; Wang L.; Teng Y.Q.; Cai M.H.; Zhao Y.H.; Nikolaev A.; Li Y.F. Determination of 123 polycyclic aromatic hydrocarbons and their derivatives in atmospheric samples. Chemosphere 2022, 296, 134025.

  1. Please justify why recovery standards (Nap-d8; Phe-d10) were analyzed in different runs and on different instruments than the original air samples. Please indicate if you corrected the results for recoveries of surrogates.

We have incorporated your comments by “

2.2. Sample analysis

The analysis of nitro-PAHs and PAHs required two instrumental runs. Total 3 nitro-PAHs were detected using an Agilent 7000A GC/MS Triple Quadrupole System (Agilent Technologies Inc, USA) and quantitatively analyzed in MRM mode. Ion source: inert ion source (EI). Chromatographic column: Agilent 19091J-433 HP-5 (30 m x 0.25mm x 0.25 mm). The injected volume was 1 µL in splitless mode. The parameters were as follows: inlet temperatures were 240 ◦C for nitro-PAHs, 80 oC ramp to 240 oC (at a rate of 20 oC /min) and held for 7.5 min, then ramp to 300 oC (at a rate of 30 oC /min) and held for 2.5 min; the initial temperature of GC was 80 ◦C (held 3 min) for PAHs (Naphthalene-d8 and phenanthrene-d10), and then heat to 200 oC (at a rate of 10 oC /min) and held for 4 min, last heat to 300 oC (at a rate of 6 oC /min) and held for 8 min. Due to various isomers were contained in detected nitro-PAHs and PAHs, different oven temperature program were tried to select the optimal method, which has not only the better analytical peaks but also the shorter separation time [1]. The initial collision energy (CE) was set as 10–60 eV to determine the characteristic fragment ions. Lastly, CE was further optimized with the increase in sensitivity under Multi Reaction Monitor (MRM) mode. Different CE parameters were selected from 5 eV to 60 e V and the other parameters were same, appropriate CE parameter was determined by observing the corresponding signal conditions of each target peak for 3 nitro-PAHs [1].

3 nitro-PAHs were separated and the analysis time was 26.397 min for 2-Nitrofluorene (CE:15 eV), 27.170 min for 9-Nitroanthracene (CE:15 eV) and 32.241 min for 1-Nitropyrene (CE:25 eV).”.

Process blank, laboratory blank and field blank were analyzed, and the measured values of PAHs and NPAHs could be ignored. Since satisfying recoveries were obtained, actual sample monitoring results were not corrected by the blank and internal standard recovery.

Round 2

Reviewer 3 Report

Dear Authors, thank you for considering and applying Reviewers' comments and suggestions; this corrected version of the manuscript reads much better.  The English style could still be improved in the corrected version.  I recommend this manuscript for publishing.  Thank you.